# Robust Training of Federated Models with Extremely Label Deficiency

**Yonggang Zhang**[1*]  **Zhiqin Yang**[1*]  **Xinmei Tian**[2]  **Nannan Wang**[3]  **Tongliang Liu**[4]  **Bo Han**[1†]

[1]TMLR Group, Hong Kong Baptist University    [2]University of Science and Technology of China
[3]Xidian University    [4]Sydney AI Centre, The University of Sydney

## Abstract

Federated semi-supervised learning (FSSL) has emerged as a powerful paradigm for collaboratively training machine learning models using distributed data with label deficiency. Advanced FSSL methods predominantly focus on training a single model on each client. However, this approach could lead to a discrepancy between the objective functions of labeled and unlabeled data, resulting in gradient conflicts. To alleviate gradient conflict, we propose a novel twin-model paradigm, called **Twin-sight**, designed to enhance mutual guidance by providing insights from different perspectives of labeled and unlabeled data. In particular, Twin-sight concurrently trains a supervised model with a supervised objective function while training an unsupervised model using an unsupervised objective function. To enhance the synergy between these two models, Twin-sight introduces a neighbourhood-preserving constraint, which encourages the preservation of the neighbourhood relationship among data features extracted by both models. Our comprehensive experiments on four benchmark datasets provide substantial evidence that Twin-sight can significantly outperform state-of-the-art methods across various experimental settings, demonstrating the efficacy of the proposed Twin-sight. The code is publicly available at: github.com/tmlr-group/Twin-sight.

## 1 Introduction

Federated learning (FL) (Yang et al., 2019; Kairouz et al., 2021; Li et al., 2021; McMahan et al., 2017; Wang et al., 2020) has gained widespread popularity in machine learning, enabling models to learn from decentralized devices under diverse domains (Li et al., 2019; Xu et al., 2021; Long et al., 2020a). Despite the benefits of FL, obtaining high-quality annotations remains challenging in resource-constrained scenarios, often leading to label deficiency and degraded performance (Jin et al., 2023). In this regard, federated semi-supervised learning (FSSL) (Diao et al., 2022; Liu et al., 2021c; Jeong et al., 2021) has achieved significant improvements in tackling label scarcity by jointly training a global model using labeled and/or unlabeled data.

Advanced FSSL methods propose to combine off-the-rack semi-supervised methods (Sohn et al., 2020; Xie et al., 2020a; Berthelot et al., 2020) with FL (McMahan et al., 2017; Li et al., 2020), leveraging the strengths of both approaches like pseudo-labeling (Lee et al., 2013) and teacher-student models (Tarvainen & Valpola, 2017). These methods typically train a single model on each client using labeled or unlabeled data, following the inspirits of traditional semi-supervised learning. However, the decentralized nature of FL scenarios distinguishes FSSL from traditional semi-supervised learning, where labeled and unlabeled data are on the same device. Namely, clients in FL may have diverse capabilities to label data, leading to label deficiency on many clients (Liu et al., 2021c; Yang et al., 2021; Liang et al., 2022). Training a single model using different objective functions could make gradients on different distributions collide, as depicted in Figure 2(a). Thus, it is urgent to develop an FL-friendly semi-supervised learning framework to tackle label deficiency.

To combat label deficiency, we propose a twin-model paradigm, called **Twin-sight**, to enhance mutual guidance by providing insights from different perspectives of labeled and unlabeled data, adapting

---

*Equal contributions.
†Correspondence to Bo Han (bhanml@comp.hkbu.edu.hk).

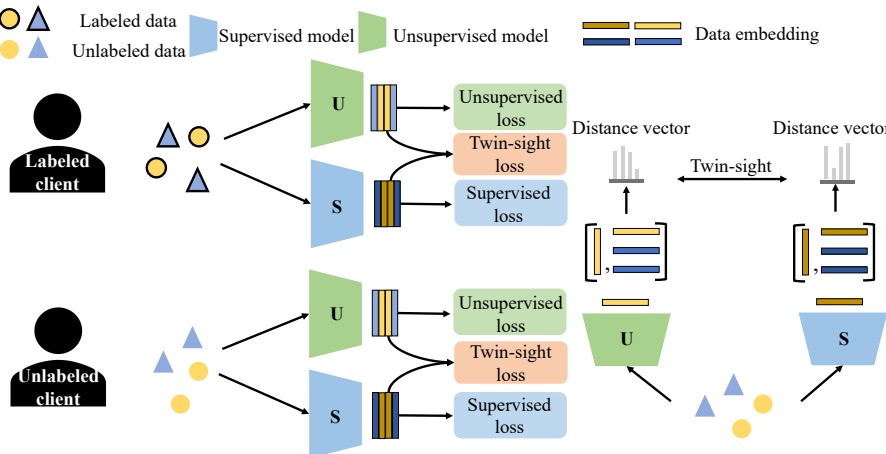

Figure 1: Overview of Twin-sight. The framework illustrates the process for both fully-labeled and fully-unlabeled clients. Each client incorporates a supervised model and an unsupervised model. The supervised model undergoes supervised learning using either ground-truth labels or pseudo labels, while the unsupervised model performs self-supervised learning. This approach enables the generation of twin sights for each sample, capturing both supervised and unsupervised perspectives. Subsequently, these two models are aligned, leveraging the complementary information.

traditional semi-supervised learning to FL. In particular, Twin-sight trains a supervised model using a supervised objective function, while training an unsupervised model using an unsupervised objective function. The twin-model paradigm naturally avoids the issue of gradient conflict. Consequently, the interaction between the supervised and unsupervised models plays a crucial role in Twin-sight. Drawing inspiration from traditional semi-supervised learning (Belkin & Niyogi, 2004) from a manifold perspective (Roweis & Saul, 2000), we introduce a neighborhood-preserving constraint to encourage preserving the neighborhood relation among data features extracted by these two models. Consequently, the supervised and unsupervised models can co-guide each other by providing insights from different perspectives of labeled and unlabeled data without gradient conflict.

The overview of the proposed Twin-sight can be found in Figure 1. In Twin-sight, the unsupervised objective function, e.g., instance discrimination (Wu et al., 2018)[1], does not vary with the presence or absence of labels for the unsupervised model. In contrast, the supervised objective function varies with the presence or absence of labels. For clients with label information, it can be a vanilla objective, e.g., cross-entropy loss. For clients without labels, Twin-sight regards predictions with high confidence as reliable labels to perform supervised learning. In Twin-sight, the constraint remains the same whether labels exist, encouraging the preservation of neighborhood relation (Sarkar et al., 2022; Gao et al., 2023; Pandey et al., 2021). Comprehensive experiments conducted on four standard datasets demonstrate the efficacy of the proposed Twin-sight.

Overall, our contributions can be summarized as follows:

- We point out that the discrepancy between the objective functions of labeled and unlabeled data could cause gradient conflict, posing specific challenges for semi-supervised learning approaches in FL scenarios.

- To tackle label deficiency, we propose a twin-model framework, Twin-sight, to tackle gradient conflict in federated learning. Twin-sight trains a supervised model paired with an unsupervised model. Meanwhile, Twin-sight introduces a constraint to make the two models co-guide each other with insights from different perspectives of labeled and unlabeled data by preserving the neighborhood relation of data features.

---

[1]The objective function can refine class-level identification into fine-grained challenges causally (Chalupka et al., 2014; Mitrovic et al., 2020).

- We conduct comprehensive experiments under various settings using widely used benchmark datasets. Our experimental results show that Twin-sight outperforms previous methods, achieving state-of-the-art performance.

## 2 RELATED WORK

**Federated Learning.** Federated learning (FL) enables distributed clients to collaboratively train a global model with privacy-preserving (Kairouz et al., 2021; Ji et al., 2023). However, the performance of federated learning typically suffers from heterogeneity in data distributions, processing capabilities, and network conditions among clients (Lin et al., 2020; Li et al., 2022; Diao et al., 2023; Zhu et al., 2022; Tang et al., 2022). One of the most popular algorithms in FL is FedAvg (McMahan et al., 2017), which aggregates parameters from randomly selected clients to create a global model and achieves convergence after several rounds of communication. A series of works, e.g., FedProx (Li et al., 2020), SCAFFOLD (Karimireddy et al., 2020), is proposed to calibrate the local updating direction. These methods implicitly assume that all clients can label data, which could be violated in many practical scenarios. Some approaches include the sharing of privacy-free information (Tang et al., 2022) or the use of protected features (Yang et al., 2023). These strategies have shown promise in achieving better performance.

**Semi-Supervised Federated Learning (SemiFL).** To relax the assumption, SemiFL (Diao et al., 2022) assumes that the server can annotate data, while clients collect data without labels. In SemiFL, selected clients generate pseudo-labels using the global model and then fine-tune the aggregated model using labeled data on the server side. Semi-supervised learning is a well-established approach that has proven to be effective in improving the performance of machine learning models by making use of both labeled and unlabeled data (Zhu et al., 2003; Zhu & Goldberg, 2009). Self-training methods (Xie et al., 2020b; Zoph et al., 2020; Liu et al., 2021b) have emerged as a popular approach for semi-supervised learning, in which a teacher model is trained on labeled data and used to generate pseudo-labels for the remaining unlabeled data. Another significant line of work is based on consistency training (Tarvainen & Valpola, 2017; Xie et al., 2020a). Apart from the above, combining these two methods is effective in achieving improved performance on various benchmark datasets, e.g., MixMatch (Berthelot et al., 2019), FixMatch (Sohn et al., 2020), and RemixMatch (Berthelot et al., 2020). However, the server may fail to collect data due to privacy concerns.

**Federated Semi-Supervised Learning (FSSL).** Advanced works assume that the some clients have labeled data (Jin et al., 2023; Liu et al., 2021c), which has garnered significant attention. One stream of research focuses on the fully-labeled clients versus fully-unlabeled clients (Liu et al., 2021c; Yang et al., 2021; Liang et al., 2022), while another body of literature studies the use of partially labeled data at each client (Long et al., 2020b; Lin et al., 2021; Wei & Huang, 2023). For instance, RSCFed (Liang et al., 2022) leverages mean-teacher on fully-unlabeled clients and sub-sample clients for sub-consensus by distance-reweighted model aggregation. This approach comes at the cost of increased communication burden. FedIRM (Liu et al., 2021c) learns the inter-client relationships between different clients using a relation-matching module. However, these methods merely train a single model on labeled and unlabeled data, causing the gradient conflict issue.

**Self-Supervised Learning.** Self-supervised learning is an increasingly popular approach to acquiring meaningful representations without needing explicit labels (He et al., 2020; Chen & He, 2021). Contrastive methods (Wu et al., 2018; Bachman et al., 2019; Misra & Maaten, 2020) have demonstrated state-of-the-art performance, which enforces the similarity of representations between two augmented views of input. One of the predominant methods, SimCLR (Chen et al., 2020), applies InfoNCE (Oord et al., 2018) loss to discriminate positive pairs from numerous negative samples. There is a work (Zhuang et al., 2021) also investigates the federated version of these unsupervised methods. Previous work shows that the instance discrimination task can be regarded as a (more challenging) fine-grained version of the downstream task (Mitrovic et al., 2020). These insightful works inspire us to introduce self-supervised learning into FSSL for processing unlabeled data.

## 3 METHODOLOGY

In this section, we present our "***Twin-sight***" framework in detail. Before that, we provide a formal definition of the studied problem (Sec 3.1) and the motivation (Sec 3.2). We then elaborate on the

twin-model paradigm (Sec 3.3), outlining the roles and training procedures of the supervised and unsupervised models. Finally, we explore the interaction between these two sights (Sec 3.4).

## 3.1 PROBLEM DEFINITION

In general, FL tends to train a global model parameterized by $\mathbf{w}$ with $K$ participants collaboratively. In other words, the objective function $\mathcal{J}(\mathbf{w})$ of the global model is composed of the local function over all participants' data distribution:

$$\min_{\mathbf{w}} \mathcal{J}(\mathbf{w}) = \sum_{k=1}^{K} \beta_k \mathcal{J}_k(\mathbf{w}_k), \tag{1}$$

where $\beta_k$ determines the weight of the $k$-th client's objective function. The $k$-th client possesses a local private dataset denoted by $\mathcal{D}_k$, drawn from the distribution $P(X_k, Y_k)$.

In FSSL, a typical scenario involves $M$ clients with fully-labeled data, while the remaining $T$ clients have unlabeled data. The set of all clients $C = \{c_k\}_{k=1}^{K}$ can be divided into two subsets, $C^L = \{c_m\}_{m=1}^{M}$ and $C^U = \{c_t\}_{t=1}^{T}$, corresponding to the clients with labeled and unlabeled data, respectively. The dataset of the $m$-th client in $C^L$ is $\mathcal{D}_m^L = \{(\mathbf{x}_m^i, y_m^i)\}_{i=1}^{N_m} \sim P(X_m, Y_m)$ and $\mathcal{D}_t^U = \{(\mathbf{x}_t^i)\}_{i=1}^{N_t} \sim P(X_t)$ denotes the dataset containing data witout annotation for $c_t$.

## 3.2 MOTIVATION

In the existing FSSL framework, the local objective function on labeled data can be formulated as:

$$\mathcal{J}_m(\mathbf{w}_m) := \mathbb{E}_{(\mathbf{x},y)\sim P(X_m.Y_m)} \ell(\mathbf{w}_m; \mathbf{x}, y), \tag{2}$$

where $(X_m, Y_m)$ is the random variable denoting the image $X_m$ and its label $Y_m$ and $\ell(\cdot)$ is the cross-entropy loss. To leverage unlabeled data, advanced methods (Liang et al., 2022; Liu et al., 2021c; Yang et al., 2021) propose to employ traditional semi-supervised earning techniques such as pseudo-labeling (Lee et al., 2013) and mean-teacher (Tarvainen & Valpola, 2017) in conjunction with a transformation function $\mathbf{T}(\cdot)$. These methods utilize a global model parameterized to utilize unlabeled data on the $t$-th client:

$$\mathcal{J}_t(\mathbf{w}_t) := \mathbb{E}_{\mathbf{x}\sim P(X_t)} f(\mathbf{w}_t; \mathbf{x}, \mathbf{T}(\mathbf{x})), \tag{3}$$

where $f(\cdot; \cdot)$ denotes a consistency constraint. Therefore, the global objective can be rewritten as:

$$\min_{\mathbf{w}} \mathcal{J}(\mathbf{w}) = \sum_{m=1}^{M} \beta_m \mathcal{J}_m(\mathbf{w}) + \sum_{t=1}^{T} \beta_t \mathcal{J}_t(\mathbf{w}). \tag{4}$$

In centralized training, this approach can achieve state-of-the-art performance. However, the objective function may cause "client drift" due to the different objective functions of clients. Specifically, all parameters will be aggregated to construct a global model, even if these models are trained with different objective functions. In practice, aggregating models with different objective functions will cause "client drift" (Wang et al., 2020). To verify the client drifts, we calculate the similarity between gradients calculated under different objective functions, i.e., Eq. 2 and Eq. 3. The results are shown in Figure 2(a), demonstrating that gradients from these two do not align well, i.e., gradient conflict.

The gradient conflict issue is inherently attributed to the decentralized nature of data. Specifically, FL models are trained on labeled or unlabeled data, leading to aggregation with models trained using different objective functions and data distributions.

## 3.3 TWIN-MODEL PARADIGM

Built upon the aforementioned analysis, we propose to introduce a twin-model paradigm to tackle gradient conflict. Intuitively, we can train a supervised model using labeled data while training an unsupervised model using unlabeled data. Consequently, the main challenge is designing an effective interaction mechanism between these two models, making these two models promote each other by providing insights from different perspectives of labeled and unlabeled data.

---

**Algorithm 1** pseudo-code of Twin-sight

---

**Server input:** communication round $R$
**Client $k$'s input:** local epochs $E$, $k$-th local dataset $\mathcal{D}^k$

   **Initialization:** all clients initialize the model $\mathbf{w}_{s,k}^0, \mathbf{w}_{u,k}^0$.
   **Server Executes:**
   **for** each round $r = 1, 2, \cdots, R$ **do**
      server random samples a subset of clients $C_r \subseteq \{1, ..., K\}$,
      server **communicates** $\mathbf{w}_s^r, \mathbf{w}_u^r$ to selected clients
      **for** each client $c_k \in C_r$ **in parallel do**
         $\mathbf{w}_{u,k}^{r+1}, \mathbf{w}_{s,k}^{r+1} \leftarrow$ Local_Training $(k, \mathbf{w}_s^r, \mathbf{w}_u^r)$
      **end for**
      $\mathbf{w}_s^{r+1}, \mathbf{w}^{r+1} \leftarrow$ AGG $(\mathbf{w}_{s,k}^{r+1}, \mathbf{w}_{u,k}^{r+1}, c_k \in C_r)$
   **end for**

   **Local_Training$((k, \mathbf{w}_s^r, \mathbf{w}_u^r))$:**
   **if** $c_k \in C^L$ **then**
      $\mathbf{w}_{u,k}^{r+1}, \mathbf{w}_{s,k}^{r+1} \leftarrow$ SGD update by Eq 10 in $E$ epochs.
   **else if** $c_k \in C^U$ **then**
      $\mathbf{w}_{u,k}^{r+1}, \mathbf{w}_{s,k}^{r+1} \leftarrow$ SGD update by Eq 11 in $E$ epochs.
   **end if**
   **Return** $\mathbf{w}_{u,k}^{r+1}, \mathbf{w}_{s,k}^{r+1}$ to server

---

The twin-model paradigm has two models: an unsupervised model parameterized with $\mathbf{w}_u$ and a supervised model parameterized with $\mathbf{w}_s$. The unsupervised model is trained with a fine-grained task of a downstream classification task, i.e., instance discrimination:

$$\min_{\mathbf{w}_u} \mathcal{J}^u(\mathbf{w}_u) = \underbrace{\sum_{m=1}^M \beta_m \mathcal{J}_m^u(\mathbf{w}_u)}_{\substack{Unsupervised\ model \\ on\ \textbf{fully-labeled}\ clients}} + \underbrace{\sum_{t=1}^T \beta_t \mathcal{J}_t^u(\mathbf{w}_u)}_{\substack{Unsupervised\ model \\ on\ \textbf{fully-unlabeled}\ clients}}, \tag{5}$$

where the objective function $\mathcal{J}^u(\cdot)$ is the same for all client[2]:

$$\mathcal{J}^u(\mathbf{w}_u) = -\log \frac{\exp\left(\operatorname{sim}\left(f(\mathbf{w}_u; \mathbf{x}_i), f(\mathbf{w}_u; \mathbf{x}_j)\right)/\tau\right)}{\sum_{k=1}^{2N} \mathbb{I}_{[k \neq i]} \exp\left(\operatorname{sim}\left(f(\mathbf{w}_u; \mathbf{x}_i), f(\mathbf{w}_u; \mathbf{x}_k)\right)/\tau\right)}, \tag{6}$$

where $f(\mathbf{w}_u; \cdot)$ is the unsupervised model and $\tau$ is the temperature hyper-parameter. Thus, the unsupervised model can be trained in a vanilla FL manner.

Supervised models on clients with labeled data can be trained with a cross-entropy loss $\mathcal{J}_m(\cdot)$. Notably, the label information is invalid on clients sampled from the unlabeled subset $C^U = \{c_t\}_{t=1}^T$. Thus, we introduce a surrogate loss $\mathcal{J}_t^s(\cdot)$ to train the supervised model with unlabeled data on client $c_t$. This can be formulated as:

$$\min_{\mathbf{w}_s} \mathcal{J}^s(\mathbf{w}_s) = \sum_{m=1}^M \beta_m \mathcal{J}_m(\mathbf{w}_s) + \sum_{t=1}^T \beta_t \mathcal{J}_t^s(\mathbf{w}_s), \tag{7}$$

where the surrogate loss replaces the label used in cross-entropy loss with a pseudo label $\tilde{y}$ predicted by the supervised model $f(\mathbf{w}_s; \cdot)$:

$$\mathcal{J}_t^s(\mathbf{w}_s) := -\mathbb{I}\left[\sigma(\tilde{y}) > r\right] \sigma(\tilde{y}) \log f(\mathbf{w}_s; \mathbf{x}_i), \tag{8}$$

where $\mathbb{I}(\cdot)$ is an indicator function, $\sigma(\cdot)$ can select the maximum value for a given vector, and $r$ is a threshold working as a hyper-parameter to select predictions with high confidence. This is because training models using data with low-confidence predictions cause performance degradation, which is consistent with previous work (Wang et al., 2022). Consequently, we can train a supervised model in a vanilla FL manner.

---

[2] Here, we omit the difference induced by distribution discrepancy between clients.

### 3.4 TWIN-SIGHT INTERACTION

Training two models separately cannot make these two models benefit each other. Therefore, we introduce a Twin-sight loss to complete the Twin-sight framework. The inspiration is drawn from local linear embedding (Roweis & Saul, 2000) and distribution alignment (Zhang et al., 2022), where the features (or embeddings) of the same data should keep the same neighborhood relations under different feature spaces.

Specifically, we introduce a constraint to encourage preserving the neighborhood relation among data features extracted by supervised and unsupervised models. The intuition is straightforward that features extracted by the supervised model and the unsupervised model can be drastically different, making it hard to align the feature distributions. Thus, we propose to Twin-sight loss $\mathcal{J}_a(\cdot)$ to align the neighborhood relation among features:

$$\min_{\mathbf{w}_s, \mathbf{w}_u} \mathcal{J}_a(\mathbf{w}_s, \mathbf{w}_u) := d(\mathbf{N}(f(\mathbf{w}_s; \mathbf{x})), \mathbf{N}(f(\mathbf{w}_u; \mathbf{x})), \tag{9}$$

where $d$ is a certain metric to measure the difference between two matrices, e.g., $\ell_F$-norm and $\mathbf{N}(\cdot)$ stands for the function used to construct a neighborhood relation. The Twin-sight loss can be used to train both the supervised and unsupervised models in a vanilla FL manner. Consequently, the objective function on labeled data $\mathcal{J}^l(\cdot)$ is formulated as:

$$\mathcal{J}^l(\mathbf{w}_s, \mathbf{w}_u) = \underbrace{\mathcal{J}_m(\mathbf{w}_s)}_{\substack{Supervised\ model \\ on\ \textbf{fully-labeled}\ clients}} + \underbrace{\lambda_u \mathcal{J}^u(\mathbf{w}_u)}_{\substack{Unsupervised\ model \\ on\ \textbf{fully-labeled}\ clients}} + \underbrace{\lambda_d \mathcal{J}_a(\mathbf{w}_s, \mathbf{w}_u)}_{\substack{Twin-sight\ interaction \\ on\ both\ models}}, \tag{10}$$

Similarly, we can leverage unlabeled data by loss function $\mathcal{J}^u(\cdot)$:

$$\mathcal{J}^u(\mathbf{w}_s, \mathbf{w}_u) = \underbrace{\mathcal{J}^s_t(\mathbf{w}_s)}_{\substack{Supervised\ model \\ on\ \textbf{fully-unlabeled}\ clients}} + \underbrace{\lambda_u \mathcal{J}^u(\mathbf{w}_u)}_{\substack{Unsupervised\ model \\ on\ \textbf{fully-unlabeled}\ clients}} + \underbrace{\lambda_d \mathcal{J}_a(\mathbf{w}_s, \mathbf{w}_u)}_{\substack{Twin-sight\ interaction \\ on\ both\ models}}, \tag{11}$$

where $\lambda_u$ and $\lambda_d$ is the hypter-parameters to adjust.

The overview of the Twin-sight framework is illustrated in Figure 1 and Algorithm 1. According to the framework of Twin-sight, it is also possible to apply Twin-sight to a similar label deficiency scenario where all clients hold data with a portion of it labeled. This superiority is supported by our experiments, as shown in Table 3.

## 4 EXPERIMENTS

To evaluate our method, we have structured this section into four parts: 1) Detailed description of the datasets and baseline methods used in this paper within FSSL (Sec 4.1). 2) The main results that demonstrate the efficacy of our proposed method (Sec 4.2). 3) Extensive evaluations of Twin-sight to another scenario in FSSL, where all clients possess partially labeled data (Sec 4.3).

### 4.1 EXPERIMENTAL SETUP

**Datasets.** In our experiments, we use four popular datasets that have been extensively utilized in FSSL research (Liang et al., 2022; Wei & Huang, 2023) including CIFAR-10 (Krizhevsky et al., 2009), SVHN (Netzer et al., 2011), Fashion-MNIST (FMNIST) (Xiao et al., 2017), and CIFAR-100 (Krizhevsky et al., 2009). The training sets of these four datasets are $50,000$, $73,257$, $60,000$, and $50,000$ respectively. They are partitioned into $K$ clients in federated learning, and we resize all images to $32 \times 32$ size.

**Federated Semi-supervised Learning Setting.** 1) *Data heterogeneity:* To simulate data heterogeneity, we partition the dataset across clients using the Latent Dirichlet Sampling (LDA) strategy (Hsu et al., 2019), with $\gamma$ in $Dir(\gamma)$ controlling the label and quantity skewness of the data distribution among clients. In our experiments, we mainly use a severe non-IID setting with $\gamma = 0.1$ (Fig. 2(b) (a) shows the data distribution across 10 clients ), which closely resembles real-world scenarios and is important for evaluating the effectiveness of federated learning algorithms. 2) *FSSL:* We follow the setting of existing FSSL works (Liang et al., 2022). Specifically, our federated learning (FL) system

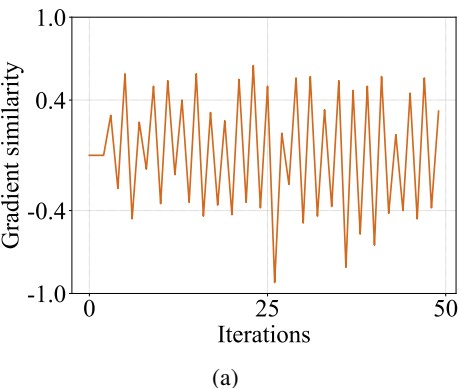 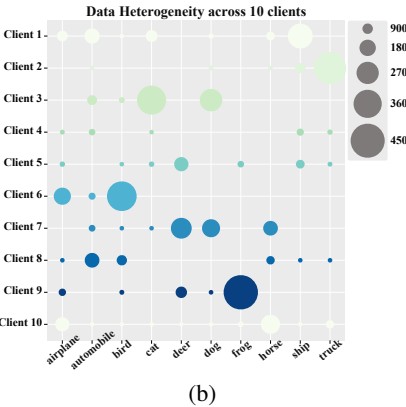

(a)                                              (b)

Figure 2: (a) The gradient similarity between two objective functions, i.e., defined on labeled and unlabeled data, throughout the training process. The figure demonstrates the gradient conflict. (b) Data heterogeneity under $Dir(\gamma = 0.1)$. Each bubble indicates the number of $y$-th class at client $k$.

comprises $K$ clients, among which $M$ have access to fully-labeled training data, and $T$ have only unlabeled data. The proportion of fully-unlabeled clients, represented by the ratio $\alpha = \frac{T}{K}$, constitutes a key factor determining the extent of annotation scarcity in Twin-sight, while $(1 - \alpha) = \frac{K-T}{K} = \frac{M}{K}$ highlights the degree of label richness across the participating clients.

**Baselines.** To verify the performance and robustness of Twin-sight, we compare it against several methods, including the combination of semi-supervised and FL methods, as well as other state-of-the-art baseline methods in FSSL. 1) FedAvg (McMahan et al., 2017), trained only with labeled data as a lower bound for comparison. 2) FedProx (Li et al., 2020), proposed to mitigate heterogeneous scenarios in FL. 3)FedAvg+FixMatch (McMahan et al., 2017; Sohn et al., 2020), the combination of two excellent methods in the respective fields of federated learning (FL) and semi-supervised learning (SSL. 4) FedProx+FixMatch (Li et al., 2020; Sohn et al., 2020), revise federated learning strategy to fit into heterogeneity. 5) FedAvg+Freematch (McMahan et al., 2017; Wang et al., 2023), vanilla FL method deployed with SOTA semi-supervised framework. 6)FedProx+Freematch (Li et al., 2020; Wang et al., 2023), a combination of two methods too. 7) Fed-Consist (Yang et al., 2021), use consistency loss computed by augmented data. 8) FedIRM (Liu et al., 2021c), a relation matching scheme between fully-labeled clients and fully-unlabeled clients. 9)RSCFed (Liang et al., 2022), randomly sub-sample for sub-consensus.

**Implementation Details.** Similar to many works (Tang et al., 2022; Wei & Huang, 2023; Huang et al., 2024), we use Resnet-18 (He et al., 2016) as a backbone feature extractor on all datasets and baselines to ensure a fair comparison. In federated learning, we aggregate weights in a FedAvg (McMahan et al., 2017) manner. In accordance with previous works (Liang et al., 2022), all of our experimental results report on the performance of the global model after $R = 500$ rounds of training. The server randomly samples a subset of all clients which means $|C_r| = 5$ when clients number $K = 10$, namely the sampling rate $S = 50\%$. The random seed in our experiments is 0. We use the SGD optimizer with a learning rate of 0.01, weight decay of 0.0001, and momentum of 0.9 in all of our experiments. The batch size is set to 64 for all datasets.

## 4.2 MAIN RESULTS

The experimental results for Twin-sight on CIFAR-10, SVHN, FMNIST, and CIFAR-100 are presented in Table 1 and Table 2. The experiments were conducted using the same random seed, with 6 out of 10 clients randomly selected to be fully-unlabeled clients while the remainder were fully-labeled clients, namely $\alpha = 60\%$, and we select 5 clients per communication round ($S = 50\%$) in FL system. Overall, the performance of both the baseline methods and Twin-sight is lower than the upper bound of FedAvg. However, our proposed method, "Twin-sight," demonstrates a significant improvement in performance, outperforming all baselines, indicating successful mitigation of the

Table 1: The performance of Twin-sight is compared to state-of-the-art (SOTA) methods on CIFAR-10 and CIFAR-100, with $\gamma = 0.1$ and $K = 10$.

| Method | No. Fully-labeled Clients/Fully-unlabeled Clients | | CIFAR-10 | | CIFAR-100 | |
|---|---|---|---|---|---|---|
| | Labeled Clients (M) | Unlabeled Clients (T) | Acc↑ | Round↓ | Acc↑ | Round↓ |
| Vanilla FL method | | | | | | |
| FedAvg-Upper Bound (McMahan et al., 2017) | 10 | 0 | 82.78 | | 64.45 | |
| FedAvg-Lower Bound | 4 | 0 | 61.58 | 295 | 48.36 | 469 |
| FedProx-Lower Bound (Li et al., 2020) | 4 | 0 | _63.66_ | _168_ | 44.64 | None |
| Combination of FL and SSL method | | | | | | |
| FedAvg+FixMatch (McMahan et al., 2017; Sohn et al., 2020) | 4 | 6 | 63.58 | 207 | _48.73_ | **315** |
| FedProx+FixMatch (Li et al., 2020; Sohn et al., 2020) | 4 | 6 | 62.44 | 269 | 43.61 | None |
| FedAvg+Freematch (McMahan et al., 2017; Wang et al., 2023) | 4 | 6 | 58.47 | None | 48.67 | 417 |
| FedProx+Freematch (Li et al., 2020; Wang et al., 2023) | 4 | 6 | 59.28 | None | 40.45 | None |
| Existing FSSL method | | | | | | |
| Fed-Consist (Yang et al., 2021) | 4 | 6 | 62.42 | 231 | 47.31 | None |
| FedIRM (Liu et al., 2021c) | 4 | 6 | – | – | – | – |
| RSCFed (Liang et al., 2022) | 4 | 6 | 60.78 | None | 43.48 | None |
| **Twin-sight (Ours)** | 4 | 6 | **70.06** | **115** | **49.98** | _400_ |

The performance of FedAvg-Lower Bound is the target accuracy. "Round" refers to the communication round required to reach the target accuracy. "None" indicates that this method did not attain the target accuracy throughout the entire training period. The bold indicates the best result, while the underlined represents the runner-up.

Table 2: The performance of Twin-sight is compared to state-of-the-art (SOTA) methods on SVHN and FMNIST, with $\gamma = 0.1$ and $K = 10$.

| Method | No. Fully-labeled Clients/Fully-unlabeled Clients | | SVHN | | FMNIST | |
|---|---|---|---|---|---|---|
| | Labeled Clients (M) | Unlabeled Clients (T) | Acc↑ | Round↓ | Acc↑ | Round↓ |
| FedAvg-Upper Bound | 10 | 0 | 87.34 | | 88.98 | |
| FedAvg-Lower Bound | 4 | 0 | 51.10 | 70 | 72.46 | 172 |
| FedProx-Lower Bound | 4 | 0 | 49.22 | None | 70.71 | None |
| FedAvg+FixMatch | 4 | 6 | 58.68 | **35** | 67.52 | None |
| FedProx+FixMatch | 4 | 6 | 45.58 | None | 63.20 | None |
| FedAvg+Freematch | 4 | 6 | _59.74_ | _45_ | 63.10 | None |
| FedProx+Freematch | 4 | 6 | 50.91 | None | 69.62 | None |
| Fed-Consist | 4 | 6 | 56.87 | 103 | 68.51 | None |
| RSCFed | 4 | 6 | 54.50 | 69 | _76.58_ | **88** |
| **Twin-sight (Ours)** | 4 | 6 | **62.94** | 125 | **79.95** | _140_ |

gradient conflict. Specifically, our method achieves excellent results on all datasets, with a particularly notable improvement on CIFAR-10.

Despite its potential advantages, RSCFed did not exhibit superior performance compared to our methods due to the presence of gradient conflict (see Figure 2(a)).However, the combination of FedAvg (McMahan et al., 2017) and Fixmatch (Sohn et al., 2020) or Freematch (Wang et al., 2023) achieved comparable performance in certain scenarios, leveraging two fundamental methods from different fields despite its simplicity. Moreover, FedIRM results in a NaN loss when used in severely skewed label distributions.

## 4.3 PARTIALLY LABELED DATA SCENARIO

Furthermore, we explore the scenario where all clients have partially labeled data. To quantify the availability of labeled data for each client, we introduce $\tau$, which represents the labeled data ratio, indicating the proportion of labeled data available. In addition to the vanilla FL method and the combination of FL and Semi-supervised learning (SSL) methods used in the previous setting, we

Table 3: The performance of Twin-sight is compared to state-of-the-art (SOTA) methods on CIFAR-10, CIFAR-100, SVHN and FMNIST in another scenario with $\gamma = 0.1$ and $K = 10$.

| Method | Labeled ratio ($\tau$) | Unlabeded data ratio | CIFAR-10 | SVHN | CIFAR-100 | FMNIST |
|---|---|---|---|---|---|---|
| Vanilla FL method | | | | | | |
| FedAvg-Upper Bound | 100% | 0% | 82.78 | 87.34 | 64.45 | 88.89 |
| FedAvg-Lower Bound | 5% | 0% | 45.35 | 37.81 | 19.46 | 75.21 |
| FedProx-Lower Bound | 5% | 0% | 45.44 | 27.34 | 20.47 | 79.77 |
| Combination of FL and SSL method | | | | | | |
| FedAvg+FixMatch | 5% | 95% | 74.97 | 64.44 | 33.58 | 75.62 |
| FedProx+FixMatch | 5% | 95% | 60.89 | 67.34 | 23.01 | **81.09** |
| FedAvg+Freematch | 5% | 95% | 75.47 | 68.43 | 44.16 | 74.78 |
| FedProx+Freematch | 5% | 95% | 64.73 | 69.01 | 31.78 | 76.75 |
| Existing FSSL method | | | | | | |
| FedSem (Albaseer et al., 2020) | 5% | 95% | 43.17 | 63.41 | 20.11 | 76.87 |
| FedSiam (Long et al., 2020b) | 5% | 95% | 47.05 | 57.18 | 21.25 | 80.53 |
| FedMatch (Jeong et al., 2021) | 5% | 95% | 52.86 | 69.08 | 23.64 | 80.16 |
| **Twin-sight (Ours)** | 5% | 95% | **78.89** | **73.24** | **45.62** | 80.11 |

incorporate three additional methods specifically designed for this partially labeled data scenario, FedSem (Albaseer et al., 2020), FedSiam (Long et al., 2020b), and FedMatch (Jeong et al., 2021). The results presented in Table 3 highlight the remarkable improvements achieved by Twin-sight in the new scenario, with the exception of the FMNIST dataset. The observed performance difference in the FMNIST dataset could be attributed to the fact that the algorithm's performance has reached a bottleneck when trained on only $5\%$ of the available data. However, Twin-sight still demonstrates comparable results with other methods on the FMNIST dataset.

## 5 CONCLUSION

In this work, we present ***Twin-sight***, a novel twin-model paradigm designed to address the challenge of label deficiency in federated learning (FL). There are three key factors contributing to the improvement of Twin-sight. First of all, we decouple the learning objective into two models which avoids gradient conflicts. In the most important part, twin-sight interaction, our unsupervised model conducts an instance classification task which is a fine-grained classification problem. Namely, this task would contribute to the downstream classification tasks (Mitrovic et al., 2020). Moreover, the data, model, and the objective function are consistent among all clients. Lastly, our supervised model conducts a classification task. Furthermore, the data, model, and objective functions are consistent among all clients, except for some unlabelled data paired with pseudo labels.

**Limitation** The twin-model paradigm introduces an additional model, which can potentially increase memory and communication overhead in federated learning (FL). As part of our future work, we aim to explore a memory-friendly dual-model paradigm that addresses these concerns.

**Future works** Currently, few existing methods can effectively address multiple FSSL scenarios. Therefore, future research should focus on proposing multi-scenario generalization and robust methods capable of handling FSSL problems in various situations. Furthermore, it is essential to consider communication overhead, computation overhead, and performance in the experimental evaluations to provide diverse solutions that cater to the different requirements of cross-silo and cross-device scenarios.

## ETHIC STATEMENT

This paper does not raise any ethical concerns. This study does not involve any human subjects, practices to data set releases, potentially harmful insights, methodologies and applications, potential conflicts of interest and sponsorship, discrimination/bias/fairness concerns, privacy and security issues, legal compliance, and research integrity issues.

## REPRODUCIBILITY STATEMENT

To make all experiments reproducible, we have listed all detailed hyper-parameters of each FL algorithm. Due to privacy concerns, we will upload the anonymous link of source codes and instructions during the discussion phase to make it only visible to reviewers.

## ACKNOWLEDGMENTS AND DISCLOSURE OF FUNDING

Yonggang Zhang, Zhiqin Yang and Bo Han were supported by the NSFC General Program No. 62376235, Guangdong Basic and Applied Basic Research Foundation No. 2022A1515011652, HKBU Faculty Niche Research Areas No. RC-FNRA-IG/22-23/SCI/04, CCF-Baidu Open Fund, and HKBU CSD Departmental Incentive Scheme. Xinmei Tian was supported in part by NSFC No. 62222117, the Fundamental Research Funds for the Central Universities under contract WK3490000005, and KY2100000117. Nannan Wang was supported in part by the National Natural Science Foundation of China under Grants U22A2096. Tongliang Liu is partially supported by the following Australian Research Council projects: FT220100318, DP220102121, LP220100527, LP220200949, and IC190100031.

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

# A EXPERIMENTAL DETAILS

## A.1 THE SYMBOLIC REPRESENTATION

In this section, we give a description of the symbol in our paper in Table 4.

Table 4: The Symbolic description.

| Symbolic representation | Description |
| --- | --- |
| $\beta_k$ | the aggregation weight of client $k$ |
| $C$ | the set of all clients |
| $M$ | the number of fully-labeled clients |
| $T$ | the number of fully-unlabeled clients |
| $K$ | the number of all clients ($|C|$) |
| $R$ | the communication round |
| $S$ | the sampling rate |
| $E$ | the local epochs |
| $\gamma$ | the non-iid degree |
| $\alpha$ | the proportion of fully-unlabeled clients |
| $\tau$ | the ratio of labeled data of each client |

## A.2 DATA VISUALIZATION

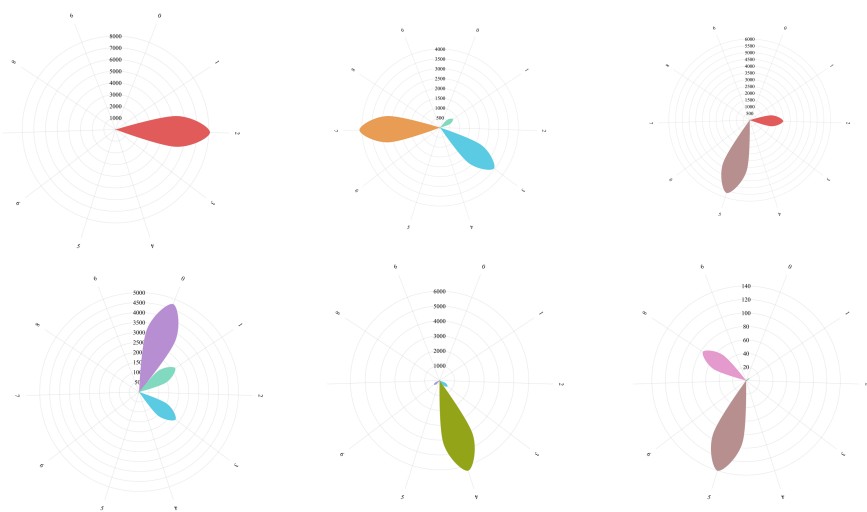

Figure 3: The data distribution of different clients under SVHN.

In this section, we present the data distribution of different clients on the SVHN dataset after performing LDA partition with a parameter value of $\gamma = 0.1$. The visualization clearly illustrates that each client predominantly contains samples from a specific class, indicating a significant concentration of data within individual client distributions. We also visualize another dataset CIFAR-10 in Figure 4, which contain both the distribution of clients and samples of fully-labeled client and fully-unlabeled client.

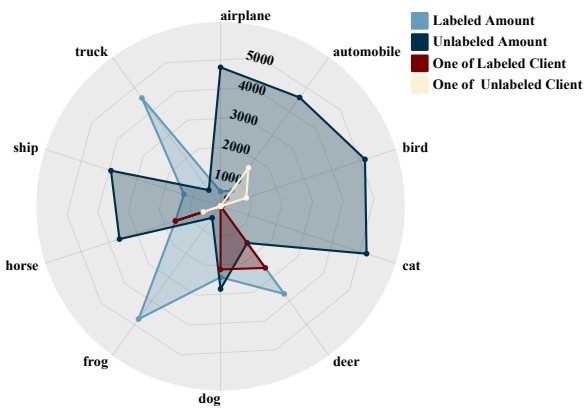

Figure 4: The blue and black areas in the figure correspond to the total amount of labeled and unlabeled data, respectively. The other two areas in the figure show the class distribution of a labeled client and an unlabeled client. This figure reveals that the challenges faced by FSSL are not limited to the problem of label scarcity, but also include the impact of data heterogeneity.

# B  MORE EXPERIMENTAL RESULTS

## B.1  HOW DIFFERENT DATA HETEROGENEITY AFFECT TWIN-SIGHT?

In this section, we aim to assess the robustness and generalization capability of FSSL under varying levels of data heterogeneity. In addition to considering the degree of $\gamma = 0.1$ in the main paper, we also explore a more severe heterogeneous scenario $\gamma = 0.05$, similar to what conventional methods have investigated in prior works (Luo et al., 2021; Li et al., 2022). In this experiment, we maintained a consistent sampling rate and communication round as in the previous experiments. Specifically, we applied a sampling rate ($S$) of 50%, which involved randomly selecting 5 clients out of the total 10 clients. Additionally, we conducted a total of 500 communication rounds ($R$) during the experiment.

Table 5: The performance of Twin-sight is compared to state-of-the-art (SOTA) methods on CIFAR-10 and SVHN, with $\gamma = 0.05$ and $K = 10$.

| Method | No. Fully-labeled Clients/Fully-unlabeled Clients | | CIFAR-10 | | SVHN | |
|---|---|---|---|---|---|---|
| | Labeled Clients (M) | Unlabeled Clients (T) | Acc↑ | Round↓ | Acc↑ | Round↓ |
| FedAvg-Upper Bound | 10 | 0 | 75.34 | | 79.13 | |
| FedAvg-Lower Bound | 4 | 0 | 35.21 | 480 | 32.26 | **165** |
| FedProx-Lower Bound | 4 | 0 | 39.02 | 323 | 33.92 | 293 |
| FedAvg+FixMatch | 4 | 6 | 40.18 | 263 | 35.32 | 405 |
| FedProx+FixMatch | 4 | 6 | 29.61 | None | 20.93 | None |
| FedAvg+Freematch | 4 | 6 | 33.98 | None | 32.68 | 361 |
| FedProx+Freematch | 4 | 6 | 33.26 | None | 24.68 | None |
| Fed-Consist | 4 | 6 | 37.19 | 301 | 31.76 | None |
| RSCFed | 4 | 6 | 38.96 | **73** | 40.16 | 354 |
| **Twin-sight (Ours)** | 4 | 6 | **43.82** | 117 | **43.22** | 367 |

The results of the experiments conducted under different data heterogeneity degrees are presented in Table 5 and Table 6. When analyzed in conjunction with the results from Table 1 and Table 2, these findings highlight the impact of data heterogeneity on the performance of the evaluated methods. As the non-iid degree ($\gamma$) decreased, indicating a more severe level of data heterogeneity among clients, the performance of all methods showed a decline. However, it is worth noting that our approach demonstrated a slower rate of performance decline compared to the other methods. These results

Table 6: The performance of Twin-sight is compared to state-of-the-art (SOTA) methods on CIFAR-100 and FMNIST, with $\gamma = 0.05$ and $K = 10$.

| Method | No. Fully-labeled Clients/Fully-unlabeled Clients | | CIFAR-100 | | SVHN | |
|---|---|---|---|---|---|---|
| | Labeled Clients (M) | Unlabeled Clients (T) | Acc↑ | Round↓ | Acc↑ | Round↓ |
| FedAvg-Upper Bound | 10 | 0 | 62.01 | | 78.34 | |
| FedAvg-Lower Bound | 4 | 0 | 41.57 | 290 | 47.10 | 255 |
| FedProx-Lower Bound | 4 | 0 | 39.41 | None | 48.5 | 403 |
| FedAvg+FixMatch | 4 | 6 | 43.19 | 222 | 43.27 | None |
| FedProx+FixMatch | 4 | 6 | 37.92 | None | 42.58 | None |
| FedAvg+Freematch | 4 | 6 | 42.96 | 273 | 42.16 | None |
| FedProx+Freematch | 4 | 6 | 35.77 | None | 41.63 | None |
| Fed-Consist | 4 | 6 | 41.67 | 379 | 40.73 | None |
| RSCFed | 4 | 6 | 43.48 | **200** | 48.5 | **159** |
| **Twin-sight (Ours)** | 4 | 6 | **43.57** | 221 | **50.6** | 176 |

suggest that our method exhibits greater robustness and resilience in the face of increasing data heterogeneity.

## B.2 HOW TO DETERMINE THE SELF-SUPERVISED METHOD?

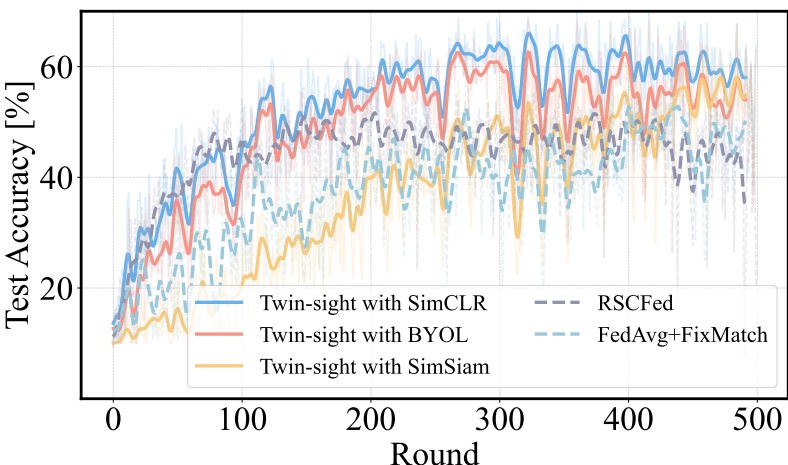

Figure 5: Different self-supervised model with our methods under the setting of $\gamma = 0.1, E = 1, K = 10$, CIFAR-10.

To verify the performance of Twin-sight with different self-supervised methods, namely BYOL and SimSiam in addition to SimCLR, we conducted experiments and analyzed the results shown in Figure 5. The results indicate that Twin-sight exhibits competitive performance regardless of the specific self-supervised method employed. While SimCLR achieved the best performance, Twin-sight still demonstrated strong performance when using BYOL or SimSiam.

This suggests that Twin-sight is adaptable to different self-supervised methods and can effectively leverage their benefits within the twin-sight framework. The ability of Twin-sight to achieve favourable results across multiple self-supervised approaches highlights its versatility and robustness in incorporating different self-supervised learning techniques.

Table 7: Comparison of Twin-sight's performance with different sampling rates $(S)$ on the CIFAR-10 dataset, under the setting of $\gamma = 0.1$ and $K = 10$ clients.

| Method | 50% | | 40% | | 30% | | 20% | |
|---|---|---|---|---|---|---|---|---|
| | Acc ↑ | Round↓ | Acc ↑ | Round ↓ | Acc ↑ | Round↓ | Acc ↑ | Round↓ |
| FedAvg-Lower Bound | 61.58 | 295 | 60.78 | 429 | 51.21 | 474 | 49.76 | 489 |
| FedProx-Lower Bound | 63.66 | 68 | 64.33 | 285 | 59.14 | **133** | 55.91 | 271 |
| FedAvg+FixMatch | 63.58 | 207 | 63.14 | 392 | 57.21 | 368 | 52.12 | 455 |
| FedProx+FixMatch | 62.44 | 269 | 60.12 | None | 56.39 | 399 | 51.79 | 398 |
| FedAvg+Freematch | 58.47 | None | 56.71 | None | 50.95 | None | 51.18 | 498 |
| FedProx+Freematch | 59.28 | None | 57.34 | None | 61.53 | 223 | 54.98 | 280 |
| Fed-Consist | 62.42 | 231 | 61.09 | 462 | 54.14 | 405 | 50.35 | 465 |
| RSCFed | 60.78 | None | 59.04 | None | 58.24 | 73 | 53.71 | 267 |
| Twin-sight (Ours) | **70.06** | **115** | **69.15** | **214** | **63.84** | 134 | **61.52** | **226** |

## B.3 WHAT WILL HAPPEN WHEN THE SAMPLING RATE $(S)$ IS SMALL?

In a real-world federated learning setting, it is common for not all clients to be continuously online due to various factors such as network connectivity issues, power constraints, or intermittent availability. To simulate this on-and-off situation, we randomly select a subset of clients denoted as $C_r$ from the complete set of clients $C = \{c_k\}_{k=1}^{K}$. In each communication round, we determine the fraction of clients to be sampled using a parameter called the sampling rate $(S)$. This sampling rate represents the proportion of clients that are randomly selected from the entire client pool $C$ for participation in that specific round.

If the available clients are small, what will happen? To investigate the scenario where only a small number of clients are available, we conducted exploratory experiments on the CIFAR-10 dataset using different sampling rates $(S)$. In this case, we reduced the sampling rate from the commonly used $50\%$ to $20\%$ among the 10 available clients with $R = 500$ communication rounds. By decreasing the sampling rate to $20\%$, we simulated a scenario where a smaller fraction of clients actively participated in each communication round. This situation reflects scenarios where federated learning is carried out with limited client availability.

The results are presented in Table 7. Upon analysis, it is evident that as the sampling rate decreases, which corresponds to a smaller number of clients being sampled in each round, there are noticeable impacts on both the overall performance and the convergence speed of the global model. However, despite the reduced client participation, our method consistently maintains a significant performance gain when compared to other baseline methods. For instance, when transitioning from a $50\%$ to a $40\%$ sampling rate, the performance degradation of our method remains below $1\%$. This indicates that even with a reduction in client availability, our method effectively mitigates the performance drop, ensuring robust results. Moreover, when the sampling rate decreases to $20\%$, our method demonstrates a sustained performance level above $60\%$, further highlighting its robustness and effectiveness.

## B.4 DOES TWIN-SIGHT REMAIN EFFECTIVE ACROSS DIFFERENT NUMBERS OF CLIENTS $(K)$?

The total number of clients in a federated learning (FL) system poses a potential challenge for Twin-sight. As the number of clients increases, each client possesses a smaller amount of data, reflecting the cross-device scenario in the simulated federation. In this setting, different devices have limited data availability, and some devices lack the capability to label data, resulting in unlabeled data for those devices.

To assess its effectiveness in scenarios with varying client numbers, we conducted experiments and evaluated the results. The outcomes are presented in Table 8 and Table 9 under CIFAR-10. To simulate a scenario with a large number of clients, we selected 50 clients, of which $60\%$ had no labeled data while the remaining clients had labeled data available. In this setup, we sampled 5 out

Table 8: The performance of Twin-sight is compared to state-of-the-art (SOTA) methods on CIFAR-10, with $\gamma = 0.1$, $K = 50$ and $\alpha = 60\%$.

| Method | No. Fully-labeled Clients/Fully-unlabeled Clients | | 50 clients | |
|---|---|---|---|---|
| | Labeled Clients (M) | Unlabeled Clients (T) | Acc ↑ | Round ↓ |
| FedAvg-Lower Bound | 20 | 30 | 16.31 | 348 |
| FedProx-Lower Bound | 20 | 30 | 32.31 | 44 |
| FedAvg+FixMatch | 20 | 30 | 41.21 | 13 |
| FedProx+FixMatch | 20 | 30 | 54.74 | **7** |
| FedAvg+Freematch | 20 | 30 | 21.52 | 345 |
| FedProx+Freematch | 20 | 30 | 37.89 | 134 |
| Fed-Consist | 20 | 30 | 36.41 | 78 |
| RSCFed | 20 | 30 | 57.96 | 14 |
| Twin-sight (Ours) | 20 | 30 | **62.2** | 13 |

Table 9: The performance of Twin-sight is compared to state-of-the-art (SOTA) methods on CIFAR-10, with $\gamma = 0.1$, $K = 100$ and $\alpha = 60\%$.

| Method | No. Fully-labeled Clients/Fully-unlabeled Clients | | 100 clients | |
|---|---|---|---|---|
| | Labeled Clients (M) | Unlabeled Clients (T) | Acc ↑ | Round ↓ |
| FedAvg-Lower Bound | 40 | 60 | 28.94 | 495 |
| FedProx-Lower Bound | 40 | 60 | 44.91 | 132 |
| FedAvg+FixMatch | 40 | 60 | 45.62 | 66 |
| FedProx+FixMatch | 40 | 60 | 54.93 | 35 |
| FedAvg+Freematch | 40 | 60 | 31.47 | 383 |
| FedProx+Freematch | 40 | 60 | 47.61 | 217 |
| Fed-Consist | 40 | 60 | 43.31 | 172 |
| RSCFed | 40 | 60 | 52.61 | **16** |
| Twin-sight (Ours) | 40 | 60 | **56.78** | 35 |

of 50 clients to participate in the FL process. Similarly, we also sampled 10 out of 100 clients to simulate a scenario with a larger client pool ($S = 10\%$).

In the scenario with 50 clients from Table 8, our method manages to achieve a performance of 62.2. Despite the reduced amount of data per client, our method demonstrates its effectiveness in leveraging the available labeled data and effectively utilizing the unlabeled data from devices that lack labeling capabilities. However, we can observe from Table 9, the reduction in client data further exacerbates the impact on performance. The limited amount of data available for each client poses a significant challenge, potentially affecting the overall performance of the system.

### B.5 HOW DOES THE PERFORMANCE OF TWIN-SIGHT VARY WITH DIFFERENT RATIOS OF UNLABELED CLIENTS?

To evaluate the robustness of our approach, we conducted experiments with varying ratios of unlabeled clients, ranging from $\alpha = 90\%$ to $\alpha = 40\%$. The results, illustrated in Figure 6, demonstrate that the ratios of unlabeled clients have an impact on the performance of Twin-sight. However, our method

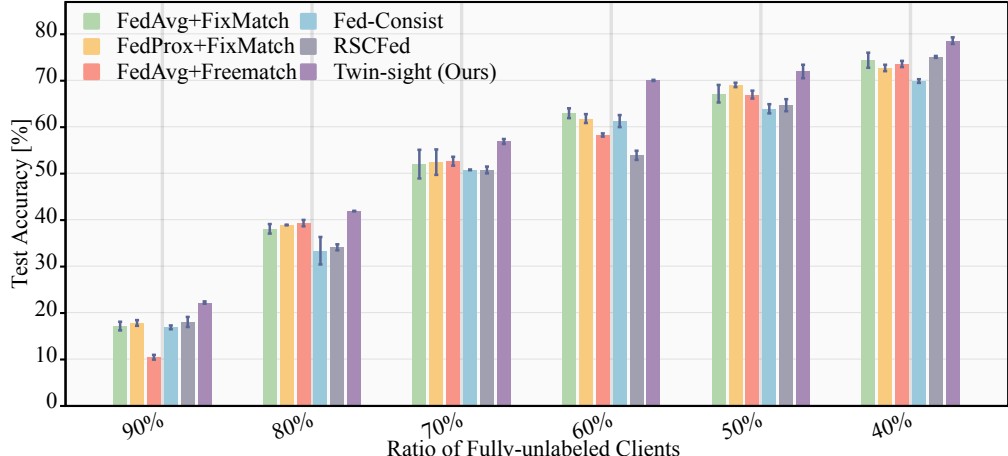

Figure 6: The effects of various ratios of unlabeled clients ($\alpha$). Comparison between five baseline methods and Twin-sight.

consistently performs well and surpasses baseline methods in these scenarios. We observed that as the number of unlabeled clients increases, the overall performance tends to degrade. Despite this, Twin-sight continues to outperform other methods even in the presence of a higher ratio of unlabeled clients. Furthermore, we noticed that Twin-sight achieves larger performance gains compared to other methods when the number of unlabeled clients is relatively small. This highlights the importance of labeled data in guiding the twin-sight interaction and improving overall performance. More experimental and ablation results can be found in Appendix B.

## B.6 DOES THE RANDOM SEED AFFECT ROBUSTNESS?

Table 10: The mean and deviation of performance with different random seeds.

| Method | CIFAR-10 | CIFAR-100 | SVHN | FMNIST |
|---|---|---|---|---|
| FedAvg-Lower Bound | $60.14 \pm 1.50$ | $48.05 \pm 0.52$ | $56.36 \pm 5.06$ | $73.32 \pm 3.79$ |
| FedProx-Lower Bound | $\underline{63.03 \pm 1.00}$ | $45.42 \pm 0.68$ | $56.03 \pm 5.90$ | $72.31 \pm 4.73$ |
| FedAvg+FixMatch | $63.87 \pm 0.73$ | $48.84 \pm 0.70$ | $56.29 \pm 2.32$ | $67.25 \pm 2.79$ |
| FedProx+FixMatch | $61.61 \pm 0.72$ | $43.55 \pm 0.46$ | $47.71 \pm 3.79$ | $65.57 \pm 2.08$ |
| FedAvg+Freematch | $58.40 \pm 2.01$ | $\underline{49.52 \pm 0.74}$ | $56.77 \pm 2.58$ | $64.78 \pm 4.67$ |
| FedProx+Freematch | $59.43 \pm 1.38$ | $41.73 \pm 0.51$ | $51.69 \pm 2.41$ | $71.68 \pm 2.51$ |
| Fed-Consist | $62.79 \pm 0.93$ | $47.94 \pm 0.70$ | $53.74 \pm 2.91$ | $66.73 \pm 3.41$ |
| RSCFed | $60.11 \pm 0.97$ | $45.90 \pm 1.32$ | $\underline{58.46 \pm 3.81}$ | $\underline{76.34 \pm 1.09}$ |
| **Twin-sight (Ours)** | $\mathbf{70.02 \pm 0.77}$ | $\mathbf{50.15 \pm 0.43}$ | $\mathbf{63.16 \pm 0.30}$ | $\mathbf{78.85 \pm 1.04}$ |

In this section, we aim to investigate the impact of different random seeds on the overall performance of the federated system. Random seed is essential for model initialization as well as client selection in each round, both of which can influence the final results. To assess the robustness of Twin-sight in the face of such variability, we select three different random seeds for each experiment and conduct tests accordingly. These experiments are conducted under $\gamma = 0.1, K = 10$ with sampling rate $S = 50\%$. The results are reported in Table 10. Notably, even when considering three random seeds, Twin-sight consistently outperforms other baseline methods across all four datasets.

### B.7 CAN INCREASING THE NUMBER OF COMMUNICATION ROUNDS ($R$) IMPROVE PERFORMANCE?

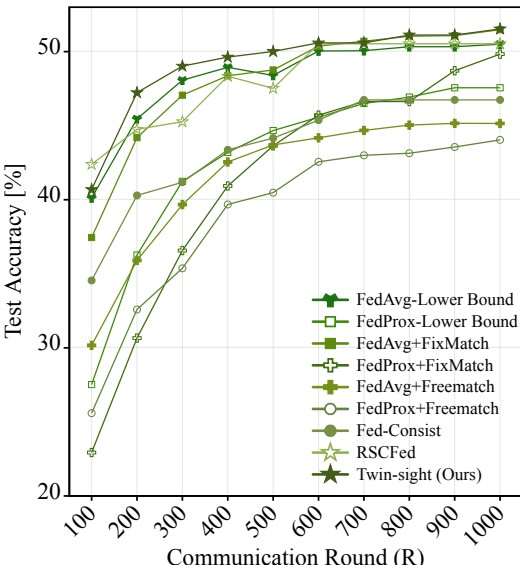

Figure 7: The performance of different communication rounds in the scenarios of fully-labeled clients and fully-unlabeled clients under CIFAR-100, with $\gamma = 0.1, K = 10$ and $\alpha = 60\%$.

The performance of a federated system is jointly influenced by the number of communication rounds $R$ and the sampling rate $S$, as they determine the necessary communication bandwidth. To evaluate the effects of different communication round settings on performance improvement, we conducted experiments in two distinct scenarios. In one scenario, a subset of clients possessed fully-labeled data, while in the other scenario, all clients had partial labels for their data. The primary objective was to achieve a balance between enhancing performance and managing communication volume effectively.

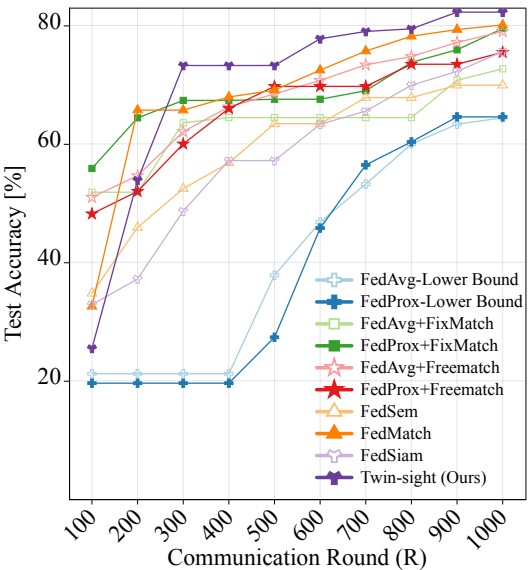

Figure 8: The performance of different communication rounds in the scenarios of all clients hold data with a portion of it labeled under SVHN, with $\gamma = 0.1, K = 10$ and $\tau = 5\%$.

As depicted in the Figure 7 and Figure 8, it is self-evident that when the number of rounds is relatively small, there is a substantial improvement in performance with each increment of 100 rounds. However, beyond 500 rounds, the performance improvement for most methods becomes less pronounced. Consequently, all the experiments presented in our paper were conducted within a maximum of 500 rounds. Our method exhibits rapid convergence and achieves excellent performance within the first 300 rounds. This observation demonstrates that Twin-sight not only enhances performance but also accelerates model convergence and reduces the number of required communication rounds. After 500 rounds, both FedAvg-Lower Bound and FedProx-Lower Bound exhibit substantial performance improvements (in Figure 8). This improvement can be ascribed to the slower convergence speed of the two lower-bound methods in this dataset, primarily due to the limited availability of labeled data. As the iterations progress, the performance gradually improves, and significant enhancements become evident after several hundred rounds of iterations.

## C    DETAILED EXPLANATIONS OF TWIN-SIGHT

In this section, we give more details and explanations of our method. Specifically, we introduce the gradient conflict problem in our paper, the self-supervised method we investigate and the twin-sight interaction.

**Gradient conflict v.s. client drift:** Gradient conflict is defined as the inconsistency between gradients. The inconsistency may result from multi-objectives, e.g., multi-task learning (MTL) (Yu et al., 2020; Liu et al., 2021a; Wang & Tsvetkov, 2021). The objective of MTL is the average loss of all kinds of tasks which typically leads to gradient conflict. According to the definition of gradient conflict, two gradients are considered conflicting if they point away from one another, i.e., have a negative cosine similarity. This phenomenon also can be observed in Figure 2(a) in our paper. The gradient conflict phenomenon has some drawbacks: i) this different loss may have various scales of gradient, and the largest one dominates the update direction; ii) The averaged objective can be quite unfavourable to a specific task's performance (Liu et al., 2021a). Client drift is caused by data heterogeneity while gradient conflict is typically induced by multiple objectives. In existing FSSL methods, the "client drift" and "gradient conflicts" often exist at the same time. In the context of label deficiency, gradient conflict may be attributed to both the samples and objectives. Specifically, partial samples have annotations and the objective on lableled and unlableled data are typically different. Advanced methods in FSSL typically use two different objective functions to train the local classifier, one for fully-labeled clients with cross-entropy loss and another for fully-unlabeled clients by unsupervised method. In this context, these methods suffer from gradient conflicts naturally.

**The self-supversied method:** In twin-sight, the $\mathcal{J}^u(\mathbf{w}_u)$ in Eq.(6) refers to the unsupervised model parameterized by $\mathbf{w}_u$. And the $\text{sim}\left(f(\mathbf{w}_u; \mathbf{x}_i), f(\mathbf{w}_u; \mathbf{x}_j)\right)$ denote the dot product between $\ell_2$ normalized embedding of data $i$ and $j$. To investigate different unsupervised methods in twin-sight, we perform some prevailing unsupervised methods, such as SimCLR (Chen et al., 2020), BYOL (Grill et al., 2020) and SimSiam (Chen & He, 2021), and SimCLR is chosen to be the backbone of $\mathcal{J}^u(\mathbf{w}_u)$. The results of different self-supervised methods are reported in Appendix B.2.

**The function $\mathbf{N}(\cdot)$ and metric $d(\cdot)$:** The function $\mathbf{N}(\cdot)$ is leveraged to quantify the relationship between different samples in a mini-batch. Specifically, $\mathbf{N}(\cdot)$ is the matrix calculating the distance among samples in the feature space, i.e., outputs of supervised model $Z_s \in \mathbb{R}^{n \times d}$ and those of unsupervised model $Z_u \in \mathbb{R}^{n \times d}$ with $d$ being the dimension and $n$ the batch size. Using these outputs, we can employ the matrix $M \in \mathbb{R}^{n \times n}$ to represents the relationships among samples, i.e., $\mathbf{N}(f(\mathbf{w}_s, \mathbf{x})) := M_s = Z_s \cdot Z_s^T, \quad \mathbf{N}(f(\mathbf{w}_u, \mathbf{x})) := M_u = Z_u \cdot Z_u^T$. Meanwhile, the metric $d(\cdot)$ is realized as the mean square error, which is formulated as: $d(\mathbf{N}(f(\mathbf{w}_s, \mathbf{x})), \mathbf{N}(f(\mathbf{w}_u, \mathbf{x}))) = \|M_s - M_u\|^2$.

