# OpenReview forum: "Robust Training of Federated Models with Extremely Label Deficiency"
_ICLR.cc/2024/Conference — ICLR 2024 poster_

### Official Review · Reviewer_37Jm · 2023-10-16

**Soundness:** 3 good
**Presentation:** 4 excellent
**Contribution:** 3 good
**Rating:** 6
**Confidence:** 3

**Summary:**

This paper points out that the objective functions used in federated semi-supervised learning cause gradient conflict, which is attributed to the decentralized nature of federated learning. To tackle gradient conflict, it proposes a dual-model framework, Twinsight, to tackle label deficiency in federated learning. Twinsight trains a supervised paired with an unsupervised model. Meanwhile, Twinsight introduces a constraint to enable the interaction between the supervised and unsupervised models to preserve the neighborhood relation.

**Strengths:**

1. The motivation of this paper is meaningful, as it raises the gradient conflict problem.

2. The proposed solution is reasonable, and preserving the proximity relationships between samples is indeed a valid approach to address this issue.

3. This paper is well-written and easy to read.

**Weaknesses:**

1. The algorithm requires training an additional model, leading to increased computation and communication costs.

2. There are details missing in the method and experimental description, raising concerns about reproducibility. Regarding Equation (9), the authors did not explicitly specify the choices of the metric d and the function N in both the method and the experiments section.

3. The datasets are basic, and it appears that each experiment was run only once, with the random seed not being disclosed.

**Questions:**

What's the specific selection of d and N in the experiments?

---

> ### Author Response · Authors · 2023-11-18
>
> We sincerely thank the reviewer for taking the time to review. We appreciate that you find that our paper has **meaningful motivation**, introduces a reasonable solution and the paper is well written. According to your valuable comments, we provide detailed feedback below and also add them to our revised paper. We hope that our response could address your concerns. We believe that these changes would improve the overall quality of our work.
>
> > Question 1: There are details missing in the method and experimental description, raising concerns about reproducibility. Regarding Equation (9), the authors did not explicitly specify the choices of the metric $d$ and the function $N$ in both the method and the experiments section. What's the specific selection of d and N in the experiments?
>
> **Answer 1:** Thanks for pointing out this problem. According to your kind suggestion, we have added the following explanations to our revision.
>
> - The function $N(\cdot)$ is used to quantify the relationship among samples. Specifically, $N(\cdot)$ is a matrix that calculates the distances among samples in the feature space, i.e., outputs of supervised model $Z_s \in \mathbb{R}^{n \times d}$ and those of unsupervised model $Z_u \in \mathbb{R}^{n \times d}$ with $d$ being the dimension and $n$ the batch size. Using these outputs, we can employ the matrix $M \in \mathbb{R}^{n\times n}$to represent the relationships among samples, i.e., $N(f(\mathbf{w}_s,\mathbf{x})):=M_s = Z_s \cdot Z_s^T, \quad N(f(\mathbf{w}_u,\mathbf{x})):=M_u = Z_u \cdot Z_u^T$.
>
> - The metric $d(\cdot)$ is realized as the mean square error, which is formulated as:
> $d(N(f(\mathbf{w}_s,\mathbf{x})), N(f(\mathbf{w}_u,\mathbf{x}))) = \|| M_s - M_u\||^2$.
>
> > Question 2: The datasets are basic, and it appears that each experiment was run only once, with the random seed not being disclosed.
>
> **Answer 2:** Thanks for your constructive comments. The random seed we use in our experiments is 0. Following your constructive question into account, we conducted the other two runs with different random seeds for both the baselines and twin-sight. The mean and deviation results are listed in the following:
>
> |                | CIFAR-10 | CIFAR-100 |  SVHN  | FMNIST |
> | -------------- | -------- | --------- | ------ | ------ |
> | FedAvg-Lower Bound |   $60.14\pm1.50$  |   $48.05\pm0.52$    |  $56.36\pm5.06$   |   $73.32\pm3.79$   |
> | FedProx-Lower Bound |   $63.03\pm1.00$  |   $45.42\pm0.68$    |  $56.03\pm5.90$   |   $72.31\pm4.73$   |
> | FedAvg+FixMatch |   $63.87\pm0.73$  |   $48.84\pm0.70$    |  $56.29\pm2.32$   |   $67.25\pm2.79$   |
> | FedProx+FixMatch |   $61.61\pm0.72$  |   $43.55\pm0.46$    |  $47.71\pm3.79$   |   $65.57\pm2.08$   |
> | FedAvg+Freematch |   $58.40\pm2.01$  |   $49.52\pm0.74$    |  $56.77\pm2.58$   |   $64.78\pm4.67$   |
> | FedProx+Freematch |   $59.43\pm1.38$  |   $41.73\pm0.51$    |  $51.69\pm2.41$   |   $71.68\pm2.51$   |
> | Fed-Consist |  62.79 $\pm0.93$  |   $47.94\pm0.70$    |  $53.74\pm2.91$   |   $66.73\pm3.41$   |
> | RSCFed |   $60.11\pm0.97$  |   $45.90\pm1.32$    |  $58.46\pm3.81$   |   $76.34\pm1.09$   |
> | Twin-sight (Ours) |   $70.02\pm0.77$ |   $50.15\pm0.43$    |  $63.16\pm0.30$   |   $78.85\pm1.04$   |
>
> We'll supplement more experiments and attach these results to our revision.
> > Question 3: The algorithm requires training an additional model, leading to increased computation and communication costs.
>
> **Answer 3:** Thanks for indicating this shortage of twin-sight. Twin-sight exactly increases the communication and computation overheads. Following your valuable comments, we have added a detailed explanation of the extra overhead induced by our method.
> Specifically, our method doubles the training time and the communication cost, because we introduce an extra model.

---

> ### Author Response · Authors · 2023-11-20
> **Welcome for more discussions**
>
> Dear reviewer #37Jm,
>
> Thank you for your valuable time reviewing our paper and providing insightful comments. We greatly appreciate your feedback, and in response to your comments, we provide thorough responses and add these valuable comments to our revision accordingly. We understand you are busy, so we have prepared **a summary of our response** for your convenience:
> - (1) **Detail of $N(\cdot)$ and metric $d(\cdot)$:** Following your valuable comment, we add the explanations of the function $N(\cdot)$ and metric $d(\cdot)$. $N(\cdot)$ is a matrix that calculates the distances among samples in the feature space. The metric $d(\cdot)$ is realized as the mean square error.
> - (2) **Experimental issue:** We have included the random seed used in our paper and have conducted two additional experiments with different random seeds. The mean and standard deviation of these experiments are provided in the table. In our revision, we will also include results from three other random seeds to enrich our results further.
>
> We sincerely hope that our response has adequately addressed your concerns. We would greatly appreciate any further feedback from you. Your insights will be invaluable in helping us further enhance the quality of our work.
>
> Best regards,
>
> Authors of #5123

---

> > ### Comment · Reviewer_37Jm · 2023-11-22
> >
> > Thank you for your response. It addressed my concerns, and I will maintain my scores.

---

> > > ### Author Response · Authors · 2023-11-23
> > >
> > > Thanks for your efforts on our paper and valuable comments. We are happy that our responses solve your concerns. Thanks a  lot!

---

> ### Author Response · Authors · 2023-11-21
> **The window for response and draft updating is closing**
>
> Dear Reviewer #37Jm,
>
> Thanks very much for your time and valuable comments. We understand you're busy, and thus, we summarize our response.
>
> We appreciate the opportunity to get your feedback, but the window for response is closing. Thus, we sincerely look forward to hearing from you, and we are very glad to provide answers following your comments.
>
> Best regards and thanks,
>
> Authors of #5123

---

### Official Review · Reviewer_NJbh · 2023-10-28

**Soundness:** 2 fair
**Presentation:** 3 good
**Contribution:** 2 fair
**Rating:** 6
**Confidence:** 3

**Summary:**

In this paper, the authors propose a novel twin-model paradigm for the federated learning in the label deficiency scenario, aimed to enhance the federated model effect by interacting insights of supervised and unsupervised models on labeled and unlabeled data. This paper points out that different objective functions of labeled and unlabeled data could lead to gradient conflict of federated learning. The author introduces supervised and unsupervised models in each client, and uses constraints to preserve the neighborhood relationship between the two models to interact their information, and finally realizes robust and effective federated semi-supervised learning. In addition, this paper conducts comparative experiments and ablation experiments on several benchmark datasets to prove the superiority and stability of the proposed algorithm compared with other baselines.

**Strengths:**

1. This paper proposes to train both supervised and unsupervised models on the client side to avoid gradient conflict caused by different objective functions when aggregating models on the server.
2. This paper introduces a neighborhood-preserving constraint to enable the supervised model and unsupervised model to fully interact with effective insights, collaborate and mutually benefit from each other’s strengths.
3.This paper conducts comprehensive comparison experiments and ablation experiments to prove the advanced performance of the proposed algorithm.

**Weaknesses:**

1. This paper combines existing techniques to solve the federal semi-supervision problem, but lacks the necessary references and explanations. For example, the unsupervised algorithm adopted in formula (6) is not explained. f(w_u;∙) as an unsupervised model, what are its specific working principles and objectives? What does sim(∙) mean? What kind of neighborhood relation construction function is N(∙) of formula (9), and how does it construct neighborhood relation?
2. The experimental results of this paper lack the necessary convincing. It is mentioned in the experimental setup that the paper will report the results after 500 rounds of training of the global model. What is the meaning of Round in Tables 1 and 2? If Round represents the number of training rounds, then why is the Round of each comparison method different and none reaches 500, which may cause the baseline results not being comparable.

**Questions:**

1. The authors can introduce the content of the proposed algorithm in more detail. For the parts that refer to other methods (in formula (6) and formula (9)), the authors should indicate the citation and briefly describe how them work.
2. The authors can rereport experimental results and compare baseline performance with the same number of training rounds.
3. In the scenario of both unlabeled and labeled data, some combinations of federated algorithms and semi-supervised algorithms does not perform as well as federated learning using only labeled data, which violates normal cognition. The authors can try to explain the reasons for this phenomenon.
4. The authors can try to prove the global convergence of the proposed objective function.

---

> ### Author Response · Authors · 2023-11-18
>
> We sincerely thank the reviewer for taking the time to review. According to your valuable comments, we provide detailed feedback below and also add them to our revised paper. We hope that our response could address your concerns. We believe that these changes would improve the overall quality of our work.
>
> > Question 1: This paper combines existing techniques to solve the federal semi-supervision problem, but lacks the necessary references and explanations. For example, the unsupervised algorithm adopted in formula (6) is not explained. $f(w_u;\cdot)$ as an unsupervised model, what are its specific working principles and objectives? What does $sim(\cdot)$ mean? What kind of neighbourhood relation construction function is $N(\cdot)$ of formula (9), and how does it construct neighborhood relation? For the parts that refer to other methods (in formula (6) and formula (9)), the authors should indicate the citation and briefly describe how them work.
>
> **Answer 1:** Thanks for your detailed comments and we apologize for the missing references and detailed explanations. Accordingly, we have added the following explanations to the revised paper.
>
> - For the unsupervised methods in Twin-sight, the $f(\mathbf{w}_u;\cdot)$ refers to the unsupervised model parameterized by $\mathbf{w}_u$. We performed some prevailing self-supervised methods, like SimCLR[R1], BYOL[R2], and SimSiam[R3] and SimCLR is choosen to be the backbone of $f(\mathbf{w}_u;\cdot)$. The results of different self-supervised methods are reported in **Appendi B.2 How to determine the self-supervised method**. And the $sim(f(\mathbf{w}_u;\mathbf{x}_i),f(\mathbf{w}_u;\mathbf{x}_j))$ denote the dot product between $\ell_2$ normalized embedding of data $i$ and $j$.
>
> - The function $N(\cdot)$ is leveraged to quantify the relationship between different samples in a mini-batch. Specifically, $N(\cdot)$ is the matrix calculating the distance among samples in the feature space, i.e., outputs of supervised model $Z_s \in \mathbb{R}^{n \times d}$ and those of unsupervised model $Z_u \in \mathbb{R}^{n \times d}$ with $d$ being the dimension and $n$ the batch size. Using these outputs, we can employ the matrix $M \in \mathbb{R}^{n\times n}$to represents the relationships among samples, i.e., $N(f(\mathbf{w}_s,\mathbf{x})):=M_s = Z_s \cdot Z_s^T, \quad N(f(\mathbf{w}_u,\mathbf{x})):=M_u = Z_u \cdot Z_u^T$.
>
> - The metric $d(\cdot)$ is realized as the mean square error, which is formulated as:
> $d(N(f(\mathbf{w}_s,\mathbf{x})), N(f(\mathbf{w}_u,\mathbf{x}))) = \|| M_s - M_u\||^2$.
>
>
> We appreciate your valuable feedback. In the updated version, we will make sure to include references and provide clear explanations to enhance the overall readability and comprehensibility.
>
> > Question 2: Lack the necessary convincing. It is mentioned in the experimental setup that the paper will report the results after 500 rounds of training of the global model. What is the meaning of Round in Tables 1 and 2? If Round represents the number of training rounds, then why is the Round of each comparison method different and none reaches 500, which may cause the baseline results not being comparable. The authors can rereport experimental results and compare baseline performance with the same number of training rounds.
>
> **Answer 2:** Thanks for your pointing out the confusing description. We have added the following explanation and fixed the problem in our revision.
>
> - In our experiments, we use $R$ to denote the total communication rounds, which is 500 in the setup part. This means we train all the methods in 500 rounds for a fair comparison.
> - In Tables 1 and 2, the 'Round' refers to the communication round required to reach the target accuracy. This can be seen in the footnote of Table 1. We use this metric to record the convergence rate of different methods, which means how many rounds the method can reach a target accuracy and 'None' denotes never reaching the target accuracy during $R=500$ rounds' training process.
>
> Thanks again for your careful review.

---

> ### Author Response · Authors · 2023-11-18
>
> >Question 3: In the scenario of both unlabeled and labeled data, some combinations of federated algorithms and semi-supervised algorithms do not perform as well as federated learning using only labeled data, which violates normal cognition. The authors can try to explain the reasons for this phenomenon.
>
> **Answer 3:** Thanks for your insightful comments. To our knowledge, some FSSL methods may lose their effect or even degenerate in a severe data heterogeneity environment. A recent work [R4] also indicates that we cannot guarantee an increase in the quantity of accurately labeled pseudo-labels during training in FSSL. For example, the pseudo-labeling technology sometimes imports many noise labels at some data distribution or dataset. These noises deteriorate the performance of the global model, leading to worse performance compared to not using the unlabeled data.
>
> >Question 4: The authors can try to prove the global convergence of the proposed objective function.
>
> **Answer 4:** Thanks for your kind suggestion. We will leave it as our future work.
>
> **Reference:**\
> [R1] Chen T, Kornblith S, Norouzi M, et al. A simple framework for contrastive learning of visual representations. In ICML, 2020.\
> [R2] Grill J B, Strub F, Altché F, et al. Bootstrap your own latent-a new approach to self-supervised learning. In NeurIPS, 2020.\
> [R3] Chen X, He K. Exploring simple siamese representation learning. In CVPR, 2021.\
> [R4] Diao E, Ding J, Tarokh V. SemiFL: Semi-supervised federated learning for unlabeled clients with alternate training. In NeurIPS, 2022.

---

> ### Author Response · Authors · 2023-11-20
> **Welcome for more discussions**
>
> Dear reviewer #NJbh,
>
> Thank you for your valuable time reviewing our paper and providing insightful comments. We greatly appreciate your feedback, and in response to your comments, we provide thorough responses and add these valuable comments to our revision accordingly. We understand you are busy, so we have prepared **a summary of our response** for your convenience:
> - (1) **Detailed explanations issues:** Following your valuable suggestions, we attach more references on the unsupervised methods. At the same time, we add detailed explanations about equations (6) and (9), the unsupervised methods, and the neighborhood function. We use the SimCLR method in our paper and calculate the distance among samples in the feature space to present $N(\cdot)$.
> - (2) **Experimental issue:** We apologize for the confusion caused by the metric 'Round.' In the response, we provide a more detailed description of "Round" and all our results exactly experiment under the same "R" for a fair comparison.
> - (3) **Anti-consensus issue:** We give the explanations that some other inappropriate approaches may instead introduce more noise labels, leading to worse results than if the unlabeled data were not utilized.
>
> We sincerely hope that our response has adequately addressed your concerns. We would greatly appreciate any further feedback from you. Your insights will be invaluable in helping us further enhance the quality of our work.
>
> Best regards,
>
> Authors of #5123

---

> > ### Comment · Reviewer_NJbh · 2023-11-20
> > **Official Comment by Reviewer NJbh**
> >
> > Thank you for your clear response. My concerns have been resolved. I have also read other comments and detailed responses, and I would like to improve my score to marginally above the acceptance threshold.

---

> > > ### Author Response · Authors · 2023-11-21
> > >
> > > We are glad that our responses addressed your questions. We want to express our gratitude for upgrading your score!

---

### Official Review · Reviewer_QrkV · 2023-10-28

**Soundness:** 3 good
**Presentation:** 3 good
**Contribution:** 3 good
**Rating:** 6
**Confidence:** 4

**Summary:**

This paper considers the semi-superved federated learning setting, when there are clients with only unlabeled data. Authors recognize a gradient conflict phenomenon, and proposed a two-model solution for the setting, one to conduct supervised learning and the other one unsupervised learning. It also proposed a neighborhood position constrain to align the features learnt by two models.

**Strengths:**

- The paper considers a practically important problem and proposes a useful and principled solution to it.
- The paper is overal well-written and easy to follow. It is well organized and the presentation is clear.
- The idea is straightforward and well-motivated, and has a certain degree of originality and significance.
- Extended empirical study is conducted.

**Weaknesses:**

- The motivation needs to be further explained.
  - Since "client drift" has been observed before, how "gradient cliff" is different from it, as the first contribution is claiming the phenonmenon?
  - When exactly will "gradient cliff" happen? Does it happen all the time when training traditional FSSL methods?
  - Since "The twin-model paradigm naturally avoids the issue of gradient conflict.", it would be nice to show how gradient similarities are improved by Twin-sight.

**Questions:**

- It is nice to consider the partially labeled scenario, which is more realistic. The proposed method can be straightforwardly applied to a more realistic scenario where each client has a different ratio of labeled data. Are there any considerable difficulties? This is only for discussion, authors need not to append experiments.
- What is the neighbourhood function used? Does it consider some graph structure?

---

> ### Author Response · Authors · 2023-11-18
>
> We sincerely thank the reviewer for taking the time to review. We appreciate that you find that we consider **a practically important problem**, and propose **a useful and principled solution** and the paper is well written. According to your valuable comments, we provide detailed feedback below and also add them to our revised paper. We hope that our response could address your concerns. We believe that these changes would improve the overall quality of our work.
>
> > Question 1: Since "client drift" has been observed before, how "gradient cliff" is different from it, as the first contribution is claiming the phenomenon?
>
> **Answer 1:** Thanks for pointing out this problem. According to your kind suggestion, we have added the following explanations to our revision.
>
> - Client drift is caused by data heterogeneity while gradient conflict is typically induced by multiple objectives. In existing FSSL methods, the "client drift" and "gradient conflicts" often exist at the same time.
> - Claiming the phenomenon is the motivation and the (first) contribution of this work since it motivates the proposed method. We would like to note that we did overlook the significance of this point. Thus, we sincerely thank you for your in-depth comments
>
> > Question 2: When exactly will "gradient cliff" happen? Does it happen all the time when training traditional FSSL methods? It would be nice to show how gradient similarities are improved by Twin-sight.
>
> **Answer 2:** Thanks for pointing out the potentially confusing problem. Accordingly, we have added the following explanations to our revised paper.
>
> - Gradient conflict is defined as the inconsistency between gradients, i.e., have a negative cosine similarity. The inconsistency may result from multi-objectives, e.g., multi-task learning [R1, R2, R3].
> - In the context of label deficiency, gradient conflict may be attributed to both the samples and objectives. Specifically, partial samples have annotations and the objective on lableled and unlableled data are typically different. Advanced methods in FSSL typically use two different objective functions to train the local classifier, one for fully-labeled clients with cross-entropy loss and another for fully-unlabeled clients by unsupervised method. In this context, these methods suffer from gradient conflicts naturally.
>
>
> In our experiments, gradient conflict often happens during the training process of traditional FSSL methods because of inconsistent loss function among different clients, please see Figure 2-(a). In contrast, we introduce a dual-model paradigm, avoiding the gradient conflict issue.
>
> **Reference:**\
> [R1] Yu et al. Gradient surgery for multi-task learning. In NeurIPS, 2020.\
> [R2] Liu et al. Conflict-averse gradient descent for multi-task learning. In NeurIPS, 2021.\
> [R3] Wang and Tsvetkov. Gradient Vaccine: Investigating and Improving Multi-task Optimization in Massively Multilingual Models. In ICLR, 2021
>
> >Question 3: It is nice to consider the partially labeled scenario, which is more realistic. The proposed method can be straightforwardly applied to a more realistic scenario where each client has a different ratio of labeled data. Are there any considerable difficulties? This is only for discussion, authors need not append experiments.
>
> **Answer 3:** Thanks for your insightful comments.
> - Our experiments show that our method can be effortlessly applied to partially labelled scenarios. Results shown in Table 3 demonstrate the effectiveness of our method.
> - The mentioned more realistic scenario seems a promising direction for FSSL, where each client has a different ratio of labeled data. This could be regarded as the combination of partially labeled data and partially labeled clients. In this context, twin-sight can be applied in this scenario. Unfortunately, we failed to figure out such a practical scenario. We would like to explore this setting as our further work. Thanks for your enlightening advice again.
>
> >Question 4: What is the neighbourhood function used? Does it consider some graph structure?
>
> **Answer 4:** Thanks for your valuable and insightful comments. We merely consider the distance among samples (in a batch) in the feature space. We believe employing a global relation among samples using the mentioned graph structure is a promising direction. We would like to explore it in our future work. Thanks for your kind suggestion.

---

> > ### Comment · Reviewer_QrkV · 2023-11-20
> >
> > Thank you for your clear response. My concerns on "gradient cliff" and the distance function have been resolved.
> > I have also read other comments and detailed response, and would like to keep my score.

---

> > > ### Author Response · Authors · 2023-11-20
> > > **Thanks for your prompt feedback!**
> > >
> > > We are glad to hear that we have addressed your concerns—many thanks for the kind effort you put into improving the quality of our paper.

---

### Official Review · Reviewer_r5bY · 2023-10-31

**Soundness:** 3 good
**Presentation:** 3 good
**Contribution:** 3 good
**Rating:** 6
**Confidence:** 3

**Summary:**

The authors propose Twinsight, an approach to semi-supervised federated learning that addresses gradient conflicts that may result from training a single model on each client. The Twinsight method is to train two models, one fully supervised and one unsupervised. An additional loss term is added to each model's objective to couple them together. This term quantifies the discrepancy between neighborhood relationships under the action of each model on the input data. The method is demonstrated on four data sets and compared against several baseline methods. The Twinsight method demonstrated improved performance of the supervised model.

**Strengths:**

The authors explore a novel approach for leveraging unlabeled data in the federated setting and demonstrate improved model performance as a result. The range of baselines and data sets is good.

Overall the paper is well structured and well written.

**Weaknesses:**

What are gradient conflicts? In vanilla supervised training gradients are averaged over all data. Certainly there are some data points whose gradients differ in their direction. Is this a significant cause for concern? Has it been studied elsewhere? Naturally, the dissimilarity between gradients of different data points will reach an extreme as the loss nears a local minimum. Yet, to my knowledge, this has not concerned anyone.

The neighborhood relation is a foundational technical element of the authors' method. Yet, it is given almost no technical treatment. How is N(.) computed? What are the consequences of different choices of quantifying the neighborhood?

Can the authors shed some insight into why their method appears so much better than the baselines? "This constraint enables ... to collaborate and mutually benefit from each other’s strengths" is not a strong explanation. Why should the proposed coupling be better than manifold regularization or other forms of semi-supervised learning?

**Questions:**

Can you please include the default accuracy (accuracy achieved by always predicting the majority class). This should always be included when using accuracy as an evaluation metric.

---

> ### Author Response · Authors · 2023-11-18
>
> We sincerely thank the reviewer for taking the time to review. We appreciate that you find that we propose **a novel approach** and the paper is well written. According to your valuable comments, we provide detailed feedback below and also add them to our revised paper. We hope that our response could address your concerns. We believe that these changes would improve the overall quality of our work.
> > Question 1: What are gradient conflicts? In vanilla supervised training gradients are averaged over all data. Certainly, there are some data points whose gradients differ in their direction. Is this a significant cause for concern? Has it been studied elsewhere?
>
> **Answer 1:** Thanks for pointing out the potentially confusing concept. Accordingly, we have added the following explanations to our revision.
>
> - Gradient conflict is defined as the inconsistency between gradients. The inconsistency may result from multi-objectives, e.g., multi-task learning (MTL) [R1, R2, R3].
> - In the context of label deficiency, gradient conflict can be attributed to both the samples and objectives. Specifically, this occurs when only partial samples have annotations, and the objectives for labeled and unlabeled data differ. Advanced methods in FSSL typically use two different objective functions to train the local classifier, one for fully-labeled clients with cross-entropy loss and another for fully-unlabeled clients by unsupervised method. In this context, these methods suffer from gradient conflicts naturally.
> - According to the definition of gradient conflict, two gradients are considered conflicting if they point away from one another, i.e., have a **negative cosine similarity**. This phenomenon also can be observed in Figure 2-(a) in our paper.
>
> We appreciate your insightful question about the gradient conflict while answering the general question is challenging. We will try to list some related works.
>
> - Gradient conflict caused by multi-objectives [R1].
> - Gradient conflict caused by complex model architectures [R4].
>
> The studies of gradient conflict can be found in the field of multi-task learning. The objective of MTL is the average loss of all kinds of tasks which typically leads to gradient conflict. The gradient conflict phenomenon has some drawbacks: i) this different loss may have various scales of gradient, and the largest one dominates the update direction; ii) The averaged objective can be quite unfavourable to a specific task’s performance [R2].
>
> **Reference:**\
> [R1] Yu et al. Gradient surgery for multi-task learning. In NeurIPS, 2020.\
> [R2] Liu et al. Conflict-averse gradient descent for multi-task learning. In NeurIPS, 2021.\
> [R3] Wang and Tsvetkov. Gradient Vaccine: Investigating and Improving Multi-task Optimization in Massively Multilingual Models. In ICLR, 2021\
> [R4] Li et al. Eliminating Gradient Conflict in Reference-based Line-Art Colorization. In ECCV, 2022.
>
>
>
> >  Question 2: How is $N(\cdot)$ computed? What are the consequences of different choices of quantifying the neighbourhood?
>
> **Answer 2:** Thanks for pointing out the problem. Accordingly, we have added the following explanations to our revision.
>
> - $N(\cdot)$ is leveraged to quantify the relationship between different samples in a mini-batch.
> - $N(\cdot)$ is the matrix calculating the distance among samples in the feature space, i.e., outputs of supervised model $Z_s \in \mathbb{R}^{n \times d}$ and those of unsupervised model $Z_u \in \mathbb{R}^{n \times d}$ with $d$ being the dimension and $n$ the batch size. Using these outputs, we can employ the matrix $M \in \mathbb{R}^{n\times n}$to represents the relationships among samples, i.e., $N(f(\mathbf{w}_s,\mathbf{x})):=M_s = Z_s \cdot Z_s^T, \quad N(f(\mathbf{w}_u,\mathbf{x})):=M_u = Z_u \cdot Z_u^T$.

---

> ### Author Response · Authors · 2023-11-18
>
> >  Question 3: Can the authors shed some insight into why their method appears so much better than the baselines? "This constraint enables ... to collaborate and mutually benefit from each other’s strengths" is not a strong explanation. Why should the proposed coupling be better than manifold regularization or other forms of semi-supervised learning?
>
> **Answer 3:** Thanks for your insightful comments. We would like to provide three key factors contributing to the improvement of our methods.
>
> - We decouple the learning objective into two models which avoids gradient conflicts.
> - In twin-sight interaction, our unsupervised model conducts an instance classification task which is a fine-grained classification problem. Namely, this task would contribute to the downstream classification tasks [R1]. Moreover, the data, model, and the objective function are consistent among all clients.
> - Our supervised model conducts a classification task. Furthermore, the data, model, and objective functions are consistent among all clients, except for some unlabelled data paired with pseudo labels.
>
> **Reference:**\
> [R1] Mitrovic et al. Representation Learning via Invariant Causal Mechanisms. In ICLR, 2021.
> >  Question 4: Can you please include the default accuracy (accuracy achieved by always predicting the majority class).
>
> **Answer 4:** Thanks for your detailed questions. The metrics we reported are under the global test dataset on the server side with a globally aggregated model. The global test dataset consists of 1000 data points at each class which means a balanced evaluation dataset.

---

> ### Author Response · Authors · 2023-11-20
> **Welcome for more discussions**
>
> Dear reviewer #r5bY,
>
> Thank you for your valuable time reviewing our paper and providing insightful comments. We greatly appreciate your feedback, and in response to your comments, we provide thorough responses and add these valuable comments to our revision accordingly. We understand you are busy, so we have prepared **a summary of our response** for your convenience:
> - (1) **Comprehensive explanation:** Following your constructive comments, we provide more details about gradient conflicts, which are most studied in multi-task learning.
> -  (2) **Technical detail:** For reproducibility, we give the calculation of $N(\cdot)$. On the other hand, we explain why we don't use the metric of default accuracy to compare the results.
> -  (3) **Insightful issue:** We give three key factors contributing to improving twin-sight. a) Twin-sight solves the gradient conflict naturally. b)unsupervised model conducts an instance classification task, a fine-grained classification problem. Furthermore, the data, model, and the objective function are consistent among all clients. c) supervised model conducts a classification task. Furthermore, the data, model, and objective functions are consistent among all clients, except for some unlabelled data paired with pseudo labels.
>
> We sincerely hope that our response has adequately addressed your concerns. We would greatly appreciate any further feedback from you. Your insights will be invaluable in helping us further enhance the quality of our work.
>
> Best regards,
>
> Authors of #5123

---

> ### Author Response · Authors · 2023-11-21
> **The window for response and draft updating is closing**
>
> Dear Reviewer #r5bY,
>
> We sincerely thank you for your time and valuable comments.
>
> We understand you're busy, and thus, we summarize our response. We appreciate the opportunity to get your feedback, but the window for response is closing. Thus, we sincerely look forward to hearing from you, and we are very glad to provide answers following your comments.
>
> Best regards and thanks,
>
> Authors of #5123

---

> > ### Comment · Reviewer_r5bY · 2023-11-22
> > **Author responses**
> >
> > I thank the authors for their detailed responses. I think they have clarified a number of points. I will stand by my score.

---

> > > ### Author Response · Authors · 2023-11-22
> > >
> > > We are glad that we have addressed your concerns. Thanks again for your valuable comments.

---

### Author Response · Authors · 2023-11-18
**Response to All Reviewers**

We sincerely appreciate all reviewers' great efforts in reviewing and commenting on our work. We especially thank the nice words:
- straightforward and well-motivated (Reviewer QrkV), motivation of this paper is meaningful (Reviewer 37Jm)
- novel approach (Reviewers r5bY, QrkV and NJbh), the solution is reasonable (Reviewer 37Jm).
- well structured and well written (Reviwers r5bY, QrkV, and 37Jm), easy to follow (Reviewers QrkV and 37Jm).
- baselines and data sets are good (Reviewer r5bY), and an extended empirical study is conducted (Reviewer QrkV).

At the same time, we apologize for any confusion that may have arisen and we have extracted a similar question to answer it in response to your valuable comments.

---

### Meta-Review · Area_Chair_z1zk · 2023-12-07

**Metareview:**

This paper proposes a novel twin-model paradigm for the federated learning in the label deficiency scenario, aimed to enhance the federated model effect by interacting insights of supervised and unsupervised models on labeled and unlabeled data. The authors point out that different objective functions of labeled and unlabeled data could lead to gradient conflict of federated learning. The method is demonstrated on four data sets and compared against several baseline methods. The Twinsight method demonstrated improved performance of the supervised model.

Overall, this paper receives four positive rating. The reviewers agree that this paper is overall well-written, the proposed solution is reasonable and the experimental results are good. Most of the reviews acknowledge that the motivation of this paper is clear. Based on the reviews, I lean towards acceptance.

**Justification For Why Not Higher Score:**

There are some concerns regarding to the motivation (by reviewer QrkV), experimental results (by reviewer NJbh) and computational cost (by reviewer 37Jm).

**Justification For Why Not Lower Score:**

Please refer to metareview section.

---

### Decision · Program_Chairs · 2024-01-16

Accept (poster)